# Interplay between Intestinal Bacterial Communities and Unicellular Parasites in a Morbidly Obese Population: A Neglected Trinomial

**DOI:** 10.3390/nu14153211

**Published:** 2022-08-05

**Authors:** Jana Caudet, María Trelis, Susana Cifre, José M. Soriano, Hortensia Rico, Juan F. Merino-Torres

**Affiliations:** 1Department of Endocrinology and Nutrition, University and Polytechnic Hospital La Fe, 46026 Valencia, Spain; 2Joint Research Unit on Endocrinology, Nutrition and Clinical Dietetics, University of Valencia-Health Research Institute La Fe, 46026 Valencia, Spain; 3Parasite & Health Research Group, Area of Parasitology, Department of Pharmacy and Pharmaceutical Technology and Parasitology, University of Valencia, 46010 Valencia, Spain; 4Food & Health Lab, Institute of Materials Science, University of Valencia, 46980 Valencia, Spain; 5Area of Microbiology, Department of Microbiology and Ecology, University of Valencia, 46010 Valencia, Spain; 6Department of Medicine, Faculty of Medicine, University of Valencia, 46010 Valencia, Spain

**Keywords:** obesity, eukaryotic microbiota, *Blastocystis* sp., *Giardia intestinalis*, *Dientamoeba fragilis*, metabolic markers, diet

## Abstract

Obesity is an epidemic causing a metabolic health crisis. Herein, the interactions between the gut prokaryotic and eukaryotic communities, metabolic comorbidities and diet were studied. Stool samples from 56 subjects, 47 with type III obesity and 9 with type II obesity and cardiovascular risk or metabolic disease, were assessed for the richness, diversity and ecology of the bacterial gut community through metagenomics, together with the study of the presence of common unicellular eukaryote parasites (*Blastocystis* sp., *Dientamoeba fragilis* and *Giardia intestinalis*) by qPCR. Clinical information regarding metabolic comorbidities and non-alcoholic hepatic fatty liver disease was gathered. To assess the quality of the patients’ diet, each participant filled in three dietary questionnaires. The most prevalent parasite *Blastocystis* sp. (46.4%), together with *D. fragilis* (8.9%), was found to be associated with higher mean diversity indexes regarding non-colonized subjects; the opposite of that which was observed in those with *G. intestinalis* (16.1%). In terms of phyla relative abundance, with *Blastocystis* sp. and *D. fragilis*, very slight differences were observed; on the contrary, *G. intestinalis* was related to an increase in Bacteroidetes and Proteobacteria, and a decrease in Firmicutes and Actinobacteria, presenting the lowest Firmicutes/Bacteroidetes ratio. At genus level, *Blastocystis* sp. and/or *D. fragilis* was accompanied with an increase in *Lactobacillus* spp., and a decrease in *Akkermansia* spp., *Bifidobacterium* spp. and *Escherichia* spp., while *G. intestinalis* was associated with an increase in *Bacteroides* spp., and a decrease in *Faecalibacterium* spp., *Prevotella* spp. and *Lactobacillus* spp., and the highest *Bacteroides* spp./*Prevotella* spp. ratio. Participants with non-alcoholic hepatic fatty liver presented a higher Firmicutes/Bacteroidetes ratio, and those with type 2 diabetes displayed a significantly lower *Faecalibacterium* spp./*Escherichia* spp. ratio, due to an overrepresentation of the genus *Escherichia* spp. The presence of parasites was associated with variations in the richness, diversity and distribution of taxa in bacterial communities, confirming a gain in diversity associated with *Blastocystis* sp. and providing different functioning of the microbiota with a potential positive effect on comorbidities such as type 2 diabetes, insulin resistance and metabolic syndrome. Future basic and clinical studies should assess the beneficial or pathogenic effect of these eukaryotes on obese subjects and focus on deciphering whether they may imply a healthier metabolic profile.

## 1. Introduction

Human gut microbiota encompasses millions of different microbes, including bacteria, archaea, fungi, viruses, helminths and protozoans, which are harboured by the human gastrointestinal tract in a symbiotic relationship. They carry a vast collection of genes referred to as the microbiome [1] and constitute an intestinal community that is stable over time, highly dissimilar and specific between subjects. Its functions are diverse in nature, ranging from metabolic and energy harvesting [2,3], to anti-infective and immunomodulatory [4,5].

Various studies have comprehensively described the bacterial composition of the gut microbiota, both in healthy and unhealthy subjects, even though there is still a lack of thorough understanding of its interactions with the human host. It is hypothesized that modifications in the western lifestyle (e.g., the practical disappearance of helminthic and protozoan infections, the incorporation of processed foods and the generalized use of antibiotics) have contributed to changes in the composition of the intestinal communities towards a decrease in the levels of prokaryotic and eukaryotic diversity [6,7,8]. This fact has altered the bidirectional relationship between microbiota and host, and a growing number of diseases have been suggested to be linked to this phenomenon, including obesity [9,10,11,12].

Although less abundant, unicellular eukaryotic microorganisms represent an on-topic component of the gut microbiota, but their significance within the microbial ecology remains largely unknown. Their presence has not been clearly defined as pathogenic or commensal, and knowledge about their impact on the composition of microbiota and host homeostasis remains limited. *Giardia intestinalis* is one of the most prevalent and studied parasites in this area. The establishment of *G. intestinalis* has been suggested to be conditioned by the microbiota composition of the host [7], modulating susceptibility and allowing the progression to a clinically significant infection. Furthermore, numerous studies indicate that this protozoan has the potential to reconfigure the structure of the gut ecosystem and cause a dysbiotic condition that persists even after its eradication [13,14,15,16,17].

On the other hand, *Blastocystis* sp. is one of the most common intestinal parasites isolated in human faeces, usually as an incident finding. The exposure to the parasite is insufficient to develop intestinal colonization, since its clearance depends on the resident microorganisms and the interactions between them, and with the host’s immune system. Once the parasite is established, it can exist for more than ten years, thereby becoming a stable member of the gut ecosystem and modulating its structure [11,18]. However, its role as a pathogenic organism in humans remains speculative [19], since a link with intestinal pathology has not been unequivocally demonstrated. Additionally, high bacterial diversity and richness are traits of a healthy human microbiota [20], and their loss has been linked to several disorders [21]; in this sense, the presence of *Blastocystis* sp. in the gut has been associated with a higher biodiversity and better functionality of the microbiota [22,23], and could, therefore, be considered beneficial instead of harmful. Additionally, several descriptive studies reveal a protective role of this microorganism in obesity [18,21,23].

*Dientamoeba fragilis* is another protozoan that has been neglected for decades [24]. Virtually since its discovery, there has been controversy in the scientific community about its pathogenic role, since it is common to find asymptomatic parasitized subjects [25,26]. Research regarding its impact on intestinal microbial structure is scarce, and no differences in composition or diversity have been found when comparing the intestinal communities of subjects with and without the parasite [27].

According to this current state of the art, a study on morbidly obese residents in Spain was designed to further address the interactions between these three unicellular eukaryotic parasites and the richness, diversity and ecology of the intestinal bacterial community.

## 2. Materials and Methods

### 2.1. Design of the Study

This research is part of a larger single-centre and cross-sectional study conducted from August 2018 to February 2020, in obese subjects attending the Endocrinology and Nutrition Service of University and Polytechnic Hospital La Fe (Valencia, Spain).

The main aims of this study were to describe the presence of three common enteric parasites, *Giardia intestinalis*, *Dientamoeba fragilis* and *Blastocystis* sp., and analyse their occurrence with the metagenomics results of the bacterial community analysis, outlining any differences concerning richness, diversity and ecology. A secondary objective of the study was to ascertain the differences in microbiota composition with regard to metabolic comorbidities and diet composition.

Confusing factors previously described as modifiers of the composition of human gut microbiota were registered, namely: smoking status, regular use of Proton Pump Inhibitors (PPI), *Helicobacter pylori* colonization and history of a cholecystectomy.

Informed consent was obtained from all participants, after being informed about the aim of the study, risks, and implications of their participation in it, as well as the treatment and confidentiality of the data. This study was approved by the Ethics Committee for Drug Research of the University and Polytechnic Hospital La Fe, ensuring that the fundamental principles established by the Helsinki Declaration, data protection and bioethics were respected.

### 2.2. Study Population

Data from 56 subjects were gathered. Mean age was 48.1 ± 9.8 years, with 36 women and 20 men. The individuals included in the study were mostly patients with type III (n = 47) obesity, but also a few with type II (n = 9) and comorbidities of cardiovascular risk or metabolic disease were included. Fifty participants were Spanish, while the remainder were of Bulgarian (2), Colombian (1), German (1), Dominican (1) and Honduran (1) nationality.

Inclusion criteria were: (a) age: 18–65 years old; (b) Body Mass Index (BMI): >40 kg/m^2^, or >35 kg/m^2^ in coexistence with significant comorbidity; (c) no sustained weight loss with non-surgical measures. Exclusion criteria were: (a) diagnosis of oncologic disease or active oncologic process in the last 5 years; (b) autoimmune or chronic gut inflammatory diseases; (c) history of intestinal intervention that may alter the intestinal secretion dynamics; and (d) consumption of any oral or parenteral antibiotic during the six weeks prior to faecal sampling.

Every participant put on a standard hypocaloric diet, defined as a caloric input between 20–25 kcal/kg total weight/day [28]. Additionally, patients who were close to bariatric surgery were encouraged to progressively introduce commercial hypocaloric-hyperproteic meal replacements to enhance weight loss before surgery. To this end, either Optisource^®^ (Nestlé Health Science, Vevey, Switzerland) or Vegestart Complet^®^ (Vegenat Health Care, Badajoz, Spain) products were prescribed.

### 2.3. Clinical Interview

Every patient had interviews with an endocrinologist and a nutritionist, in which relevant information was recorded regarding medical history, epidemiological data and regular drug consumption. To assess the quality of the diet followed during the previous months, each participant completed the following tools: (i) a short 14-item questionnaire of adherence to the Mediterranean diet (validated in a Spanish population and used by the PREDIMED group) [29,30]; (ii) a validated Spanish food frequency questionnaire [31]; (iii) a 24 h recall questionnaire conducted in quintuplicate, according to the European recommendations [32] and methodology [33]. The evaluation of the nutritional composition of meals and the estimation of the daily intake of energy, macro- and micronutrients was carried out using the DIAL 3.0.0.5 software [34]. The cut-offs were applied as a reference to consider whether the participants fulfilled the daily nutritional requirements or not [35].

Furthermore, the patients were categorized according to the following metabolic comorbidities: hypertension, type 2 diabetes (T2D) or pre-diabetes, dyslipidaemia and hyperuricemia. Metabolic syndrome was diagnosed according to the National Cholesterol Education Program (NCEP) Adult Treatment Panel III (ATP III) definition [36]. An insulin cut-off point of >3.8, previously validated in our population [37], was established as indicative of insulin resistance. The presence of hepatic steatosis suggestive of Non-Alcoholic Hepatic Fatty Liver Disease (NAFLD) was assessed by means of ultrasonography.

Finally, anthropometric measurements were obtained after an overnight fast to gather information concerning weight and height. A multi-frequency bioelectrical impedance analysis (InBody 770^®^, Biospace Co., Ltd., Seoul, Korea); Frequencies: 1, 5, 250, 500 and 1000 kHz) [38] and a wall height rod were employed for every participant. Body Mass Index (BMI) was calculated as kg/m^2^.

### 2.4. Stool Sampling, DNA Extraction and Parasitological Analysis

Each patient provided three stool samples collected on alternate days in REAL MiniSystems with Total-fix fixative (Durviz, Valencia, Spain) for conservation and concentration by centrifugation. Stool DNA was extracted from 200 µL of the concentrate with the QIAamp DNA Stool Mini Kit (QIAGEN, Hilden, Germany) according to the manufacturer’s instructions. Extracted DNA was stored at −20 °C until its use in parasitological and microbiological analysis.

Multiplex PCR for the detection and identification of intestinal parasites was performed using a commercially available kit, *Allplex GI-Parasite Assay* (GIPPA) (Seegene, Seoul, South Korea) for common human unicellular parasites such as *Giardia duodenalis*, *Entamoeba histolytica*, *Cryptosporidium* spp., *Blastocystis hominis*, *Dientamoeba fragilis* and *Cyclospora cayetanensis* [39,40,41]. Amplification was performed on the CFX96 Real Time PCR System (Bio-Rad, Marnes-la-Coquette, France), and managed with CFX Manager IVD 1.6 software, in a 25 µL reaction volume containing 20 µL reaction mix (5 µL Primers Mom, 5 µL Anyplex PCR MM (EM2), 8 µL of DNase/RNase-free water and 2 µL of internal control) and 5 µL of DNA. Negative (DNase/RNase-free water) and positive controls were included in each assay. Results were analysed and interpreted using the Seegene Viewer V3 software optimized for multiplex assays. Samples were considered positive for specific parasites if the cycle threshold (Ct) was ≤43 cycles, according to the manufacturer’s instructions.

### 2.5. Microbiota Analysis by Next Generation Sequencing and Interpretation

Metagenomics analysis of the intestinal bacterial community was obtained from stool DNA as described in the previous section, which was delivered to the Sequencing and Bioinformatics Service of the Foundation for the Promotion of Health and Biomedical Research of Valencia Region (FISABIO).

For the study of the intestinal microbiome, the bacterial gene coding for the *16S* rDNA was analysed according to the Illumina MiSeq *16S* Metagenomics Sequencing Library Preparation protocol (Cod. 15044223 Rev. A). The pair of primers used for the PCR was suggested by Klindworth et al. (2013) [42]. Bacterial genomic DNA (5 ng/μL in 10 mM Tris pH 8.5) was used to initiate the protocol. After *16S* rDNA gene amplification, the multiplexing step was performed using Nextera XT Index Kit (FC-131-1096). One μL of the PCR product was run on a Bioanalyzer DNA 1000 chip to verify the expected size (~550 bp). After size verification, the libraries were sequenced using a 2 × 300 pb paired-end run (MiSeq Reagent kit v3 (MS-102-3001)) on a MiSeq Sequencer according to the manufacturer’s instructions (Illumina, San Diego, CA, USA). Quality assessment was performed using the “prinseqlite” program [43], applying the following parameters: min_length: 50; trim_qual_right: 30; trim_qual_type: mean; trim_qual_window: 20.

Meta-taxonomy assessment was performed using some of QIIME2 (Quantitative Insights into Microbial Ecology) plugins. Denoising, paired-ends joining and chimera depletion were performed, starting from paired ends data using DADA2 pipeline [44]. Taxonomic affiliations were assigned using the Naïve Bayesian classifier integrated in QIIME2 plugins. The database used for this taxonomic assignation was the SILVA_release_132 [45]. The sequence data were analysed using QIIME2 pipeline, as originally cited in Caporaso (2010) [46]. The sequences are available in the ENA public repository (https://www.ebi.ac.uk/ena/browser/view/PRJEB51819 accesion number: ERA10729632) (accessed on 25 March 2022).

Microbiome diversity is typically described in terms of within (i.e., alpha) and between samples (i.e., beta) diversities. Alpha diversity of the microbiota from each sample was determined by means of Chao1 and Shannon indexes. The Chao1 index assesses the bacterial richness within a community, whereas the Shannon index evaluates the diversity in a community based on the distribution of taxa abundances within a sample. Beta diversity was analysed with Jaccard and Sorensen indexes, the most widely used in ecology and evolution, which express the similarity or dissimilarity of microbial community composition between samples. Likewise, Principal Coordinate Analyses (PCoA) were plotted against samples as a summary of the beta diversity.

Secondly, the different taxonomic categories were explored, describing the following relative abundances and ratios: (i) relative abundance of the human main intestinal phyla (Firmicutes, Bacteroidetes, Proteobacteria and Actinobacteria); (ii) Firmicutes/Bacteroidetes ratio; (iii) relative abundance of genera described as beneficial [47,48,49,50,51]: *Akkermansia* spp., *Faecalibacterium* spp., *Roseburia* spp., *Bifidobacterium* spp., and *Lactobacillus* spp.; (iv) *Bacteroides* spp./*Prevotella* spp. ratio, whose relative abundances are usually inversely correlated [52,53]; and (v) *Faecalibacterium* spp./*Escherichia* spp. ratio, which has been previously used as a dysbiosis index caused by parasitic colonization [14].

### 2.6. Statistical Analysis

Data obtained are shown as absolute frequency (%) in qualitative variables and as mean ± standard deviation or median (Q1, Q3) in quantitative variables. To compare the differences between groups, the Chi2 test for qualitative variables and Wilcoxon rank sum test with continuity correction for quantitative variables were performed. A *p* value < 0.05 was considered to indicate statistical significance when comparing clinical variables. All analyses were performed using the statistical software R (4.0 version) [54].

### 2.7. Ethical Statement

Informed consent was obtained from all participants after being informed about the aim of the study, risks and implications of their participation in it, as well as the treatment and confidentiality of the data.

This study was approved by the Biomedical Research Ethics Committee of University and Polytechnic Hospital La Fe on 1 July 2019 (Project identification code: 2017/0486), respecting the fundamental principles of the Declaration of Helsinki, of the Council of Europe Convention in relation to Human Rights and Biomedicine of the UNESCO Declaration.

## 3. Results

### 3.1. Studied Population Description

In terms of the WHO classification [55], mean BMI of the participants was 45.6 ± 6.5 kg/m^2^ and mean waist circumference 129.9 ± 13.5 cm. Active smoking was present in 25.0% of the participants, and 14.3% regularly consumed PPI. Five patients (8.9%) had a history of cholecystectomy. Up to 22 (42.3%) were taking at least one meal replacement per day during the month before faecal sampling.

The most frequent metabolic comorbidities were dyslipidaemia (73.2%), hypertension (53.6%), hyperuricemia (46.4%) and T2D (33.9%). Metabolic syndrome (ATP-III diagnostic criteria) was diagnosed in 51.7% of cases; 61.2% were insulin resistant, and 71.2% presented hepatic steatosis (US assessment).

Thirty-one participants (55.4%) were demonstrated to be colonized for intestinal unicellular parasites by multiplex qPCR. The most frequent species were *Blastocystis* sp. (46.4%), *Giardia intestinalis* (16.1%) and *Dientamoeba fragilis* (8.9%), with 5 cases combining *Blastocystis* sp./*G. intestinalis*, 2 cases of *Blastocystis* sp./*D. fragilis* and 1 case colonized by the three species. No cases of *Cryptosporidium* spp., *Entamoeba histolytica*, or *Cyclospora cayetanensis* species included in the same multiplex assay was detected.

The group of subjects colonized by a unicellular eukaryote (n = 31) was comparable with the non-colonized group (n = 25) in terms of age, gender, metabolic comorbidities, meal replacement consumption and smoking condition (Table 1). However, colonized participants were significantly more obese (mean BMI 47.3 vs. 43.4 kg/m^2^; *p* = 0.025).

### 3.2. Metagenomics Analysis of the Intestinal Bacterial Community

#### 3.2.1. Reads by Sample

Among all stool samples, the mean number of raw sequences found was 188,847. After the quality assessment protocol included in the DADA2 pipeline (initial quality filtering, denoising, paired-ends joining and chimera filtering), the mean number of reads was reduced to 136,840 (73.1% of non-chimeric inputs). The final dataset produced 3783 Amplicon Single Variants (ASV), distributed into 13 phyla, 96 families, 330 genera and 668 known species. Appendix A show individual results regarding reads by count clustered by phylum (Appendix A) and genus (Appendix A).

A trend towards a higher number of reads was found in subjects colonized with parasites, which resulted as statistically significant in the subgroup of the *Blastocystis* positives (mean number of reads: 136,282 vs. 118,070; *p* = 0.038). In addition, a significantly lower recount of sequences was identified among participants suffering from Non-Alcoholic Hepatic Fatty Liver Disease (NAFLD) (mean number of reads: 118,374 vs. 140,395; *p* = 0.002).

#### 3.2.2. Alpha and Beta Diversity

Chao1 and Shannon indexes were used at genus level to assess the intestinal bacterial structure in terms of alpha diversity. When comparing these indexes between subjects with or without intestinal parasites, higher alpha diversity was observed in *Blastocystis* sp. and/or *D. fragilis* positive ones, compared to those colonized by *G. intestinalis* (Table 2) (Figure 1); however, the differences were not significant.

The same trend was observed when analysing beta diversity (Jaccard and Sorensen indexes) and significant differences in the community structure were detected (Figure 1), with a higher variability of genera among participants colonized by *Blastocystis* sp. and/or *D. fragilis*, and lower in those colonized by *G. intestinalis* (Table 2). The divergence on the structure of the bacterial community when comparing *Blastocystis* sp. and/or *D. fragilis* cases and non-colonized subjects is reflected in Appendix A, which shows the PCoA charts of both conditions.

Evidence points to an unhealthier microbiome in subjects suffering from metabolic syndrome [20,56]. Therefore, for a more complete analysis of the alpha diversity, patients were grouped according to two variables: the colonization with parasites, and the presence of metabolic syndrome. In this combined study, the higher values of microbial diversity were found among metabolically healthy participants who harboured *Blastocystis* sp. and/or *D. fragilis* (Table 3).

On the contrary, the lowest indexes corresponded to those with metabolic syndrome and *G. intestinalis* (Figure 2).

A similar combined sub-analysis was carried out, regarding suffering from NAFLD. Consistently with the previous results, the highest diversity was obtained in NAFLD-negatives harbouring *Blastocystis* sp. and/or *D. fragilis* (Table 3) (Figure 2). After clustering according to the presence of parasites and the other metabolic comorbidities, no significant differences in alpha diversity indexes were uncovered. However, a common diversity pattern associated with parasite species was identified among patients with dyslipidaemia, hepatic steatosis, insulin resistance and T2D (Appendix A) those individuals with *Blastocystis* sp. and/or *D. fragilis* showed the greatest alpha diversity, and those with *G. intestinalis* the lowest, while non-colonized subjects displayed an intermediate situation between them. Similar results in terms of beta diversity were obtained when comparing bacterial communities at genus level with regard to metabolic comorbidities and parasitic species. Significant differences existed between non-colonized subjects and those with *Blastocystis* sp. and/or *D. fragilis*, and also with *G. intestinalis*-positive ones (Table 4).

In an attempt to analyse the differences observed in bacterial diversity with the metabolic profile of the patients, the frequency of metabolic comorbidities in the patients depending on whether or not they were colonized by *Blastocystis* sp. and/or *D. fragilis* is shown in Figure 3. As can be observed in the figure, the prevalence of metabolic syndrome, insulin resistance and T2D are lower in colonized patients. On the contrary, the prevalence of hepatic steatosis and dyslipidaemia are higher in association with colonization, although these differences are not significant in any case.

#### 3.2.3. Relative Abundance of Taxa

The most abundant phyla in the studied population were: Bacteroidetes (50.0%), Firmicutes (40.1%), Proteobacteria (6.6%), Actinobacteria (1.2%) and Verrucomicrobia (1.2%). The mean Firmicutes/Bacteroidetes ratio was 0.95. For each subject enrolled, a quantitative representation of the individual relative abundance of the bacterial phyla is reported in Appendix A.

A comparative analysis regarding the presence/absence of unicellular eukaryotes was performed, obtaining similar relative abundances of the dominating phyla between both groups. Considering that dissimilar effects may have been generated by the different parasitic species, the results were reanalysed with regard to specific parasite clusters. In case of *G. intestinalis*, an increase in Bacteroidetes and Proteobacteria relative abundance was observed compared to the other situations. The opposite occurred for Firmicutes and Actinobacteria, for which a considerable decrease was observed with respect to the non-colonized population. Regarding the Firmicutes/Bacteroidetes ratio, participants with *G. intestinalis* showed the lowest value, driven by both lower abundances of Firmicutes and higher ones of Bacteroidetes. For individuals with *Blastocystis* sp. and/or *D. fragilis*, very slight differences were observed in terms of main phyla distribution with respect to negative subjects, as shown in Table 5.

When comparing the relative abundance of taxa at genus level and sub-analysing the presence of a specific parasite, significant differences were identified with respect to non-colonized subjects. *Blastocystis* sp.-positives presented higher relative abundances of *Pyramidobacter* spp. (*p* = 0.022), *Peptococcus* spp. (*p* = 0.027), *Anaerofilum* spp. (*p* = 0.037) and *Negativibacillus* spp. (*p* = 0.04), and lower abundances of *Citrobacter* spp. (*p* = 0.034), *Enorma* spp. (*p* = 0.04) and *Ezakiella* spp. (*p* = 0.04). Subjects colonized by *D. fragilis* showed a higher relative abundance of *Aggregatibacter* spp. (*p =* 0.0031), *Brachyspira* spp. (*p =* 0.0075), *Phocea* spp. (*p =* 0.012), *Butyrivibrio* spp. (*p =* 0.017), *Pseudoflavonifractor* spp. (*p =* 0.01), *Saccharimonas* spp. (*p* = 0.02) and *Adlercreutzia* spp. (*p =* 0.04), and a lower frequency of *Papilibacter* spp. (*p =* 0.008), *Paraprevotella* spp. (*p =* 0.035) and *Snegalimassilia* spp. (*p =* 0.04). Finally, the presence of *G. intestinalis* displayed a higher relative abundance of *Murdochiella* spp. (*p* = 0.0009), *Finegolia* spp. (*p* = 0.0009), *Eubacterium* spp. (*p* = 0.005), *Flavonifractor* spp. (*p* = 0.008), *Sarcina* spp. (*p* = 0.01), *Succiniclasticum* spp. (*p* = 0.02), *Paraeggerthella* spp. (*p* = 0.02), *Oscilibacter* spp. (*p* = 0.03), *Campylobacter* spp. (*p* = 0.03), *Synergistes* spp. (*p* = 0.04) and *Ezakiella* spp. (*p* = 0.04).

Regarding a few selected genera frequently analysed in the scientific literature (Table 6), it was observed that in the group colonized by *Blastocystis* sp. and/or *D. fragilis*, there was a slight increase in *Roseburia* spp. and *Lactobacillus* spp. were prominent, and a decrease in *Akkermansia* spp., *Bifidobacterium* spp. and *Escherichia* spp. was also found. The presence of *G. intestinalis* was associated with a different genus-level pattern, with an increase in *Bacteroides* spp. and a decrease in *Faecalibacterium* spp., *Prevotella* spp. and *Lactobacillus* spp.

With respect to the *Bacteroides* spp./*Prevotella* spp. and the *Faecalibacterium* spp./*Escherichia* spp. ratios (Table 7), they globally showed no significant differences between subjects with and without parasites. However, *G. intestinalis* positives presented the highest median *Bacteroides* spp./*Prevotella* spp. ratio, and positives with *Blastocystis* sp. and/or *D. fragilis* the lowest. In relation to these results, it can be stated that the presence of *G. intestinalis* may be associated with higher abundances of the genus *Bacteroides*, which is the main representative of the phylum Bacteroidetes, in accordance with the data shown in Table 5 and Table 6. In case of the *Faecalibacterium* spp./*Escherichia* spp. ratio, the presence of parasites, either *Blastocystis* sp. and/or *D. fragilis* or *G. intestinalis* increased, which was more evident for the data with *G. intestinalis*.

Considering metabolic comorbidities, subjects suffering from NAFLD displayed significantly different gut microbial profiles with higher ratios of Firmicutes/Bacteroidetes (1.4 vs. 0.8; *p =* 0.041); a lower relative abundance of Bacteroidetes (43.5 vs. 52.9; *p =* 0.031) and a higher one of Firmicutes (47.5 vs. 37.1; *p =* 0.031). All this related to a significant increase in the genus *Faecalibacterium* spp. (4.3 vs. 3.1; *p =* 0.037) and *Clostridium* spp. (0.5 vs. 0.08; *p =* 0.002), both representatives of the predominant phylum. The condition of insulin resistance was associated with a significantly lower frequency of the genus *Blautia* spp. (0.38% vs. 0.89%; *p* = 0.015) and a higher one of *Desulfovibrio* spp. (1.0% vs. 0.2%; *p* = 0.032). Participants with T2D displayed a *Faecalibacterium* spp./*Escherichia* spp. ratio that was significantly lower (*p* = 0.01), due to an overrepresentation of the genus *Escherichia* spp., along with a trend towards a lower representation of *Akkermansia* spp. (0.5% vs. 1.3%, in non-diabetics). No differences in the composition of the microbiota at genus level were detected in relation to the presence of dyslipidaemia, hyperuricemia, hypertension or metabolic syndrome.

Finally, the structure of the microbiota regarding other variables registered during the clinical interview showed some significant results. Cholecystectomy was related at phylum level to a lower relative frequency of *Proteobacteria* (*p* = 0.02), and at genus level with a higher relative presence of *Roseburia* (*p* = 0.02) and *Blautia* (*p* = 0.03) and underrepresented genera such as *Proteus* (*p* = 0.001), *Peptoclostridium* (*p* = 0.001), *Epulopiscium* (*p* = 0.001), *Enorma* (*p* = 0.01)*, Romboutsia* (*p* = 0.01,) and *Anaerostipes* (*p* = 0.02). Secondly, the individuals who were consuming meal replacements displayed a lower relative abundance of *Clostridium* (*p* = 0.007), and the presence of rare genera such as *Fusecatenibacter* (*p* < 0.001) and *Ralstonia* (*p* < 0.01).

Another variable that provided significant results was taking proton pump inhibitors regularly, with higher relative frequency of the genus *Parabacteroides* (*p* = 0.02) and overrepresentation of infrequent genera such as: *Neisseria* (*p* < 0.001), *Brachyspira* (*p* = 0.01), *Fructobacillus* (*p* = 0.01), *Peptosclostridium* (*p* = 0.01) and *Oscillospira* (*p* = 0.03). Active colonization with *H. pylori* was coupled with a lower relative abundance of the genera *Akkermansia* (*p* = 0.02), *Collinsella* (*p* = 0.02), *Coprobacter* (*p* = 0.03) and *Bifidobacterium* (*p* = 0.04). Finally, smoking condition conferred a lower relative amount of the genera *Barnesiella* (*p* = 0.01) and *Haemophilus* (*p* = 0.04).

### 3.3. Dietary Results

Considering the information gathered from dietary questionnaires and registers, adherence to the dietetic recommendations provided by the healthcare team was, globally, extremely low. Less than one-third of the subjects ingested an optimal amount of kcal (20–25 kcal/kg adjusted weight/day), the rest either exceeding or not reaching the international recommendations. Only 14.6% were considered adherent to the Mediterranean diet, determined by a score of ≥10 points obtained in the validated questionnaire. Furthermore, fulfilment of dietary recommended daily intakes was only occasional, thus implying a very poor quality of the diet among participants: over-ingestion of simple sugars was present in 95%, of lipids in 54% and of proteins in 10%, while none of the subjects met the daily recommended intake of fibres. Insufficient dietary intake was frequently present for omega-6 (54.0%), vitamin A (80.0%), vitamin B6 (66.0%), vitamin C (49.0%) and almost in every patient for vitamin E, vitamin D, copper, zinc and iron.

A comparative metagenomics analysis of the microbiota was performed regarding the following dietary variables: total caloric intake, fulfilment of the recommended intake for the three main macronutrients and fulfilment of the daily intake of micronutrient requirements. After analysing these variables regarding the presence of parasites in a combined way, no significant difference was obtained to point to an influence of diet on microbiota diversity and parasitic colonization. However, results depending on the parasitic species in those patients with a correct daily intake of macronutrient were much appreciated. With an adequate consumption of carbohydrates, greater diversity was observed among the colonized, highlighting those of *Blastocystis* sp. and/or *D. fragilis*. Additionally, it is also interesting that when fat intake is adequate, the diversity of those colonized with *G. intestinalis* is greater than expected according to the previously detected patterns (Appendix A).

Regarding the relative abundance of the bacteria of interest, a trend to a lower relative abundance of *Faecalibacterium* spp. among omega-3-deficient subjects (*p* = 0.024) and to a lower relative abundance of Actinobacteria among omega-6-deficient ones (*p* = 0.026) was detected. Furthermore, among vitamin-A-deficient subjects, a lower abundance of *Parabacteroides* spp. (*p* = 0.029), *Lactobacillus* spp. (*p* = 0.008) and *Colinsella* spp. (*p* = 0.012), together with a lower abundance of *Lactobacillus* spp. among B12-deficient ones (*p* = 0.029), was observed.

## 4. Discussion

Several studies assessing the composition and diversity of gut microbiota have been carried out, both in healthy and unhealthy obese subjects. However, it is surprising how the literature, thus far, lacks consistent data on the interactions between eukaryotic colonization and gut microbial profiles, particularly among obese subjects. In our previous studies, a high prevalence of gut colonization with common intestinal parasites (51.0%) was detected in a population of type II and type III obese subjects [56,57]. Consequently, it was believed that exploring these interactions merited further study and the present investigation was conceived, which jointly analyses the metagenomics analysis with the presence of these microorganisms in an obese population.

The relationship between unicellular eukaryotes and prokaryotes ecology and diversity has been previously assessed and recently reviewed [58]. Several researchers have demonstrated that hosting one of these parasitic species was associated with a richer and more diverse bacterial microbiome [22,23,59,60,61]. A study by Krogsgaard et al. [62] specifically evaluated this aspect in a relatively large population, assessing the presence of *D. fragilis* and *Blastocystis* sp. via molecular analysis. A high global prevalence of parasites (47.0%) and a more diverse and richer bacterial microbiome in colonized versus non-colonized subjects was observed, supporting our results. Accordingly, we found higher richness (assessed by the number of reads) among *Blastocystis*-positive subjects. Conversely, an inverse effect between the bacterial community diversity and the presence of *G. intestinalis* was stated in the study by Mejía et al. [63], and it was suggested that this relationship may depend upon the specific parasite colonizing the gut, as well as the intensity of infection it causes. According to recent articles reviewing studies on the association between single-celled intestinal parasites and microbial communities, a differential effect could be expected depending on the species considered [64]. We assessed alpha diversity indexes in our population, comparing the parasitic species independently, and although the differences were not significant, trends described previously were confirmed, such as higher alpha and beta diversity at genus level in *Blastocystis* and/or *D. fragilis*-positive samples, and lower in *G. intestinalis*-positives.

Furthermore, we analysed whether *Blastocystis* sp. and/or *D. fragilis* were associated with higher microbiota diversity in different metabolic conditions, considering the presence of metabolic syndrome and NAFLD. The presence of these parasites was coincident with higher diversities when compared with their negative counterparts. Although we could not confirm differences attributable to the colonization, this finding seems to suggest a healthier microbiome structure in agreement with some published data [22,23,58]. We have already described, in the same obese population as the one presented here, a lower frequency of insulin resistance among subjects harbouring these unicellular eukaryotes [56]. In the present research, the greatest alpha diversity was shown by subjects colonized with *Blastocystis* sp. and/or *D. fragilis*, and without metabolic syndrome. In this context, the outcomes reported may point towards a beneficial effect of the gut colonization with *Blastocystis* sp., in agreement with recent studies, which demonstrated, in an experimental model, that long-term exposition to the parasite prior to the induction of colitis promotes a faster recovery in a protective manner, with a significant reduction in inflammatory markers [65].

Secondly, the intestinal microbiota composition was analysed with regard to the presence of intestinal parasites. At phylum level, we were not able to establish clear differences between groups, which suggests that interactions with the bacterial intestinal community are determined at a lower level (genus, species), probably related to functional niche specialization, and are also dependent on other factors. At genus level, several differences were detected regarding the occurrence of a specific parasite, but they were not consistent between parasitic species and generally affected low prevalent bacterial members. It is probable that unicellular eukaryotes display a variety of effects in shaping the gut microbiome that differ widely among each kind, which could explain the discrepancies found in the effects on microbiome composition. Additionally, considering that we compared bacterial relative abundances, the differences observed between groups may have reflected modifications in the absolute levels of a specific taxonomic category, as well as changes in other components of the microbiome. Finally, numerous factors have the potential to affect the microbiome composition, allowing confounding variables to prevent us from obtaining a clear differential pattern in colonized subjects.

The presence of *G. intestinalis* in the gut ecosystem has been related to a dysbiotic condition in several studies [13,14,66,67]. Its presence was related to a significant lower *Faecalibacterium* spp./*Escherichia* spp. ratio compared to *Blastocystis*-positive individuals [14]; the same low trend was demonstrated with the presence of other pathogenic protozoa, such as *Cryptosporidium* spp. and *Cyclospora cayetanensis* [67]. However, the prevalence of *G. intestinalis* in the studied population was too low to allow us to perform significant comparisons. Therefore, further research is needed to shed more light on the significance of this parasite in the human gut ecosystem.

Regarding *Blastocystis* sp., the results obtained in previously published metagenomics studies differ widely in the relative abundances of different taxa. A consistent finding has been a microbiome less dominated by the genus *Bacteroides* spp. in subjects harbouring this parasite [13,22,58,59], which we were not able to reproduce in the present study. Additionally, some other studies reported [22,58] a higher relative quantity of *Faecalibacterium* spp. and *Roseburia* spp., two genera considered beneficial in the gut ecosystem for their short-chain-fatty-acids production potential. We could not find any difference in the relative abundance of these genera, neither of Archaea, which has also been reported as more abundant by Beghini’s group [17], who performed the largest investigation to date on the prevalence of this parasite in human samples. They were able to identify microbiome signatures (distinguishable features that were consistent across populations) in the intestinal ecology linked to the presence of *Blastocystis* sp. On the other hand, Principal Coordinate Analysis charts reflecting beta diversity of microbial communities allowed us to establish phylogenetic distances between samples when comparing the diversity of genera with regard to the presence of *Blastocystis* sp. and/or *D. fragilis* in agreement with a previous study [61].

Additionally, we analysed the metagenomics, considering several metabolic comorbidities. An inverse relationship between the abundance of *Akkermansia muciniphila* and the BMI has been broadly described in the literature, along with a direct relationship with insulin sensitivity [68,69,70,71,72]. In this study, we observed differences in the relative abundance of other genera with regard to insulin sensitivity, such as *Blautia* spp. and *Desulfovibrio* spp. Conversely, despite the amount of evidence providing pathogenic links between the development of NAFLD and the structure of gut microbiota, human studies describing its composition in this situation are scarce and sometimes contradictory [73,74]. In our population, we detected a higher relative frequency of Firmicutes (and specifically of the genera *Faecalibacterium* spp. and *Clostridium* spp.) and a lower relative frequency of Bacteroidetes, thus causing a lower Firmicutes/Bacteroidetes ratio. Similar results in subjects suffering from NASH were found by some researchers, while other studies found the opposite alteration of the ratio [75,76]. Still unknown factors probably interfere in driving the direction of this ratio and may be determinant of the development of hepatic inflammation and fibrosis.

On the other hand, cholecystectomy is known to alter the bile flow into the intestine and therefore modify the enterohepatic circulation of bile salts, which has the potential to disrupt the structure of gut microbiota. In a case–control study [77], a higher relative abundance of two bacterium species (*Blautia obeum* and *Veillonella parvula*) was detected in subjects after cholecystectomy. In our study, we also noticed a higher relative abundance of *Blautia* spp., along with differences in the abundance of Proteobacteria (previously not reported). The meaning of these changes into the microbiome composition remains to be elucidated.

Finally, from the information gathered in this is study, it was remarkable that the participants kept to a diet of extremely poor quality, despite being in an intensive nutritional re-education programme, prior to bariatric surgery. The diet followed was considered of a western pattern in every subject, thus hindering the identification of differences in the gut microbiome structure or diversity that could have been related to this variable. Globally, it strengthens the idea of obesity as a chronic disease, and reinforces the need of preventive measures, to avoid its development from its very beginning.

A strength in our study was the number of patients on whom we performed metagenomics analysis, including subjects from different social environments. Another interesting point was the faecal samples collection, since each patient delivered three samples collected on alternating days, thereby increasing the sensitivity for detecting the genes of any intestinal unicellular eukaryotic parasite. There are some limitations of the present study that warrant consideration. First, the possibility that faecal sampling may have not been completely representative of the microbiota composition from the upper gastrointestinal tract has to be considered; thus, false negative results for some intestinal parasites may have occurred. Likewise, we cannot rule out a former infection with an intestinal parasite that had been spontaneously cleared from the intestinal ecosystem at the time of study, and hence yielded a negative result in PCR analysis. This condition may have left alterations in the composition of gut microbiota, since, as described by Beatty (2016) [15], a persistent intestinal dysbiosis-induced state is possible after infection and clearance of *G. intestinalis*, causing long-term changes in microbiota composition. Finally, we are aware that the structure and function of human intestinal microbiota is altered by a myriad of factors, many of them still unknown, which may have interfered with the results of our study.

This study provides additional data of the effects of unicellular eukaryotes on the diversity and ecology of human gut microbiota on obese subjects. The presence of *Blastocystis* sp. and/or *D. fragilis* was associated with the highest values for the mean indexes analysed, for both alpha and beta diversity, while the opposite is observed in the presence of *G. intestinalis*. The richness and diversity of the microbiota detected in association with *Blastocystis* sp. and/or *D. fragilis* could be related to a healthier metabolic profile in patients, since cases of comorbidities such as metabolic syndrome, insulin resistance and T2D are less frequent among those patients.

In terms of relative abundance of taxa, differences in bacterial community composition have been evidenced regarding parasitic species; at phylum level, in the presence of *G. intestinalis*, there was an increase in Bacteroidetes and Proteobacteria along with the lowest Firmicutes/Bacteroidetes ratio, and conversely, individuals with *Blastocystis* sp. and/or *D. fragilis* displayed a pattern similar to non-colonized subjects. At genus level, *Blastocystis* sp. and/or *D. fragilis* colonization was accompanied by a prominent increase in *Lactobacillus* spp., and a decrease in *Akkermansia* spp., *Bifidobacterium* spp. and *Escherichia* spp., while *G. intestinalis* was associated with a different genus-level pattern, with an increase in *Bacteroides* spp. and a decrease in *Faecalibacterium* spp., *Prevotella* spp. and *Lactobacillus* spp., along with the highest *Bacteroides* spp./*Prevotella* spp. ratio. Considering metabolic comorbidities, subjects suffering from NAFLD displayed significantly higher ratios of Firmicutes/Bacteroidetes, and a significant increase in the genera *Faecalibacterium* spp. and *Clostridium* spp. Participants with T2D displayed a *Faecalibacterium* spp./*Escherichia* spp. ratio that was significantly lower, due to an overrepresentation of the genus Escherichia spp., along with a trend towards a lower representation of *Akkermansia* spp.

Considering the globally high prevalence of these parasites in the human gut ecosystem, research efforts should focus on further describing the interplay with the microbiota and the host immune system. Based on our findings, we recommend designing basic and clinical studies to address the possible beneficial or pathogenic effect of these eukaryotes on shaping the bacterial community of obese subjects, and to decipher whether it may imply a healthier metabolic profile. Hopefully, future evidence on this field will provide us with new opportunities for therapeutic interventions against the development of obesity and its comorbidities.

## 5. Conclusions

This study provides additional data on the effects of unicellular eukaryotes on the diversity and composition of human gut microbiota of obese subjects. The cases of *Blastocystis* sp. and/or *Dientamoeba fragilis* presented the richest and most diverse bacterial communities according to metagenomics data. No clear patterns in the relative frequency of taxa were identified in regards of parasitic status, even though the occurrence of *Giardia intestinalis* and *D. fragilis* among participants was low. Considering the globally high prevalence of these parasites within the human gut ecosystem, research efforts should focus on further describing the interplay with the microbiota and the host immune system. Based on our findings, we recommend designing basic and clinical studies to address the possible beneficial effect of *Blastocystis* sp. on shaping the bacterial community of obese subjects, and to decipher whether it may imply a healthier metabolic profile. Hopefully, future evidence on this field will provide us with new opportunities for therapeutic interventions against the development of obesity and its comorbidities.

## Figures and Tables

**Figure 1 nutrients-14-03211-f001:**
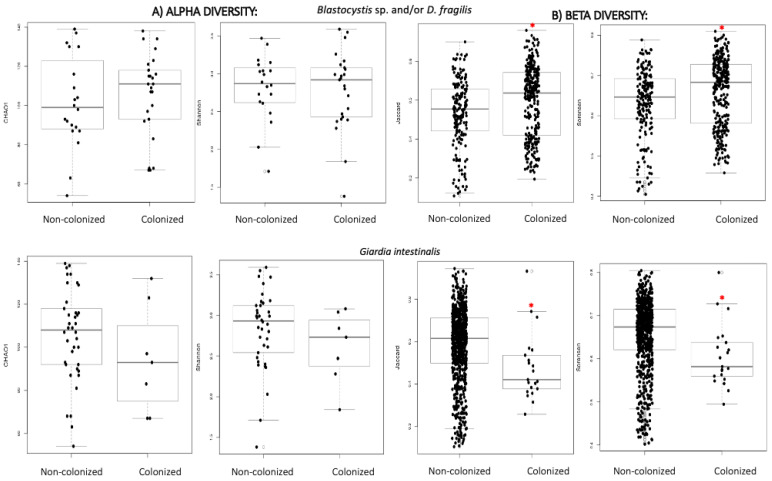
(**A**) Alpha diversity distances boxplots for comparison of genus richness (Chao1) and diversity (Shannon) between subjects regarding colonization status and parasitic species. (**B**) Beta diversity distances boxplots for comparison of microbial community composition at genus level (Jaccard and Sorensen) between the subjects clustered by colonization status and parasitic species. Horizontal lines indicate medians. * denotes *p* < 0.05 compared to the non-colonized group.

**Figure 2 nutrients-14-03211-f002:**
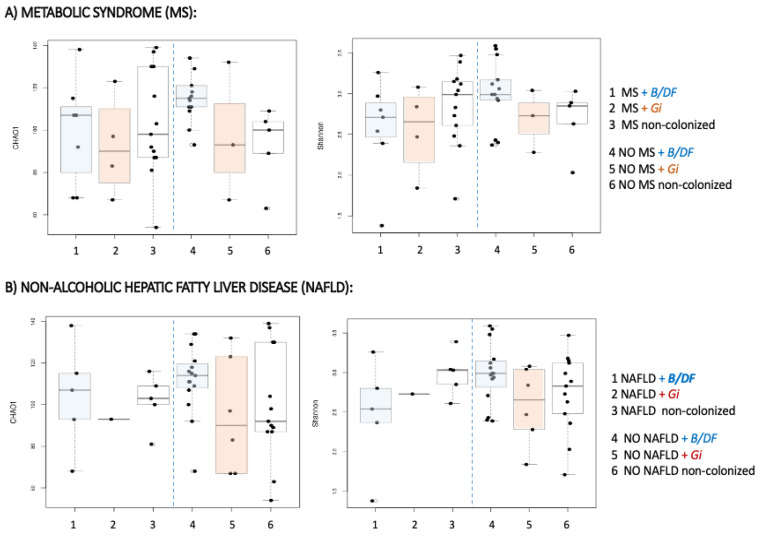
Alpha diversity distances boxplots for comparison of genus richness (Chao1) and diversity (Shannon) between subjects regarding metabolic and colonization status. (**A**) Comparison among patients regarding the colonization with parasites and the presence of metabolic syndrome. (**B**) Comparison among patients regarding the colonization with parasites and the presence of Non-alcoholic hepatic fatty liver disease. Horizontal lines indicate medians.

**Figure 3 nutrients-14-03211-f003:**
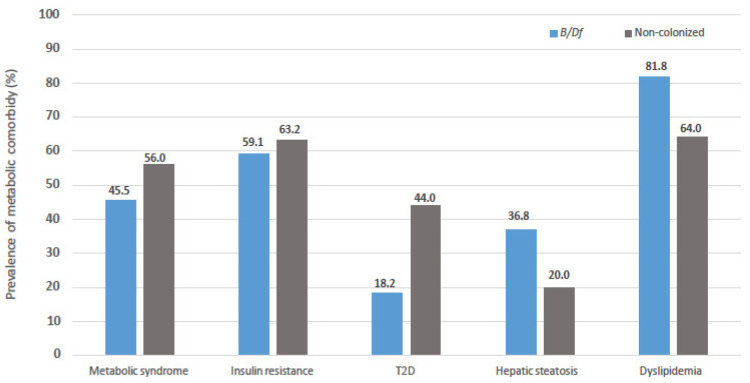
Comparison of prevalence of metabolic comorbidities in subjected enrolled regarding colonization status by *Blastocystis* sp. and/or *Dientamoeba fragilis*.

**Table 1 nutrients-14-03211-t001:** Comparison between sociodemographic, anthropometric and clinical variables categorized by colonization status.

	Colonized	Non-Colonized
***Female*** n (%)	19 (*61.3*)	17 (*68.0*)
***Male*** n (%)	12 (*38.7*)	8 (*32.0*)
***Age (years)*** mean (SD)	47.8 (*10.3*)	48.6 (*9.3*)
***BMI (kg/m^2^)*** mean (SD)	**47.3 (*6.7*)**	**43.4 (*5.8*)**
***Abdominal circumference (cm)*** mean (SD)	132.4 (*11.6*)	126.9 (*15.4*)
***Type II obesity*** n (%)	3 (*9.7*)	6 (*24.0*)
***Type III obesity*** n (%)	28 (*90.3*)	19 (*76.0*)
***Smokers*** n (%)	11 (*35.5*)	4 (*16.0*)
***Meal replacement consumption*** n (%)	11 (*35.5*)	11 (*44.0*)
***PPI regular use*** n (%)	3 (*9.7*)	5 (*20.0*)
***History of a cholecystectomy*** n (%)	1 (*3.2*)	4 (*16.0*)
***Hypertension*** n (%)	14 (*45.2*)	16 (*64.0*)
***Dyslipidaemia*** n (%)	25 (*80.6*)	16 (*64.0*)
***T2D*** n (%)	8 (*25.8*)	11 (*44.0*)
***Hepatic steatosis*** n (%)	10 ^+^ (*37.0*)	5 (*20.0*)
***Hyperuricemia*** n (%)	15 (*48.4*)	11(*44.0*)
***Metabolic syndrome*** n (%)	15 (*48.4*)	14 (*56.0*)
***Insulin resistance*** n (%)	17 (*56.7*)	12 (*63.2*)

BMI: Body Mass Index; PPI: Proton Pump Inhibitors; T2D: Type 2 Diabetes. Numbers in bold indicate significance at the 0.05 level between groups. ^+^ Missing data from 4 subjects.

**Table 2 nutrients-14-03211-t002:** Alpha and beta diversity mean indexes at genus level categorized by colonization status and parasitic species.

	Non-Colonized (n = 25)	*B/DF* (n = 22)	*G. intestinalis* (n = 9)
** *Chao 1* **	109.8	119.1	103.3
** *Shannon* **	2.89	2.94	2.72
** *Jaccard* **	0.443	0.477	0.376
** *Sorensen* **	0.599	0.632	0.520

*B/DF = Blastocystis* sp. and/or *D. fragilis* positives after excluding co-infections with *G. intestinalis*.

**Table 3 nutrients-14-03211-t003:** Mean alpha diversity indexes at genus level categorized by the presence of metabolic syndrome or NAFLD and colonization by *Blastocystis* sp. and/or *D. fragilis*.

	** *Metabolic syndrome* **		** *Metabolically healthy* **	
	Non-colonized	*B/DF*	Non-colonized	*B/DF*
** *Chao 1* **	104.9	112.5	116.3	124.5
** *Shannon* **	2.86	2.74	2.94	3.09
	** *NAFLD positives* **		** *NAFLD negatives* **	
	Non-colonized	*B/DF*	Non-colonized	*B/DF*
** *Chao 1* **	107.4	110.9	120.0	129.6
** *Shannon* **	2.90	2.75	2.89	3.09

*B/DF* = *Blastocystis* sp. and/or *D. fragilis* positives after excluding co-infections with *G. intestinalis*.

**Table 4 nutrients-14-03211-t004:** Beta diversity mean indexes at genus level regarding metabolic comorbidities and colonization.

	*Jaccard*			*Sorensen*		
	Nc	*B/DF*	*G. intestinalis*	Nc	*B/DF*	*G. intestinalis*
** *Metabolic syndrome* **	0.429	0.400	0.320 *	0.582	0.542	0.445 *
** *Dyslipidaemia* **	0.459	0.475 *	0.384 *	0.610	0.622 *	0.522 *
** *Hepatic steatosis* **	0.430	0.425	0.331 *	0.579	0.567	0.459 *
** *Insulin resistance* **	0.444	0.481 *	0.320 *	0.598	0.627 *	0.445 *
** *Type II diabetes mellitus* **	0.412	0.341	0.279 *	0.559	0.451	0.393 *

Nc= Non-colonized;* B/DF* = *Blastocystis* sp. and/or *D. fragilis* positives after excluding co-infections with *G. intestinalis*; * paired-test statistics with *p* < 0.05 when compared mean indexes of colonized with the non-colonized.

**Table 5 nutrients-14-03211-t005:** Relative abundance (%) of the main phyla clustered by colonization.

	Non-Colonized	*B/DF*	*G. intestinalis*
**Phyla**			
Firmicutes	40.1	42.2	35.0
Bacteroidetes	49.8	48.6	53.8
Proteobacteria	6.2	6.5	8.1
Actinobacteria	1.6	0.9	0.8
Verrucomicrobia	1.4	1.0	1.1
**Ratio (mean ± S.D.)**			
*Firmicutes/Bacteroidetes*	0.97 ± 0.8	1.0 ± 0.7	0.68 ± 0.2

*B/DF* = *Blastocystis* sp. and/or *D. fragilis* positives after excluding co-infections with *G. intestinalis*.

**Table 6 nutrients-14-03211-t006:** Relative abundance (%) of selected genera clustered by colonization.

	Non-Colonized	*B/DF*	*G. intestinalis*
*Faecalibacterium* spp.	3.6	3.8	2.0
*Roseburia* spp.	2.9	3.6	2.2
*Akkermansia* spp.	1.3	0.9	1.1
*Bacteroides* spp.	21.3	20.5	27.1
*Prevotella* spp.	11.5	11.0	10.5
*Lactobacillus* spp.	0.09	0.82	0.03
*Bifidobacterium* spp.	0.68	0.28	0.51
*Escherichia* spp.	0.68	0.25	0.67

*B/DF* = *Blastocystis* sp. and/or *D. fragilis* positives after excluding co-infections with G. *intestinalis*.

**Table 7 nutrients-14-03211-t007:** Comparison of main genus ratios clustered by colonization.

	Non-Colonized	*B/DF*	*G. intestinalis*
	Median [Q1, Q2]	Median [Q1, Q2]	Median [Q1, Q2]
*Bacteroides* spp./*Prevotella* spp.	460 [0.3, 11.300]	200 [0.3, 4.681]	1.625.0 [0.9, 3.687.1]
*Faecalibacterium* spp./*Escherichia* spp.	40.6 [9.0, 107.6]	63.5 [1.9, 277.8]	84.6 [0.9, 345.0]

*B/DF* = *Blastocystis* sp. and/or *D. fragilis* positives after excluding co-infections with G. *intestinalis*.

## Data Availability

The sequences generated after metagenomics are available in the ENA public repository (https://www.ebi.ac.uk/ena/browser/view/PRJEB51819 accesion number: ERA10729632) (accessed on 25 Mach 2022).

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
