# Peer review of "Interplay between Intestinal Bacterial Communities and Unicellular Parasites in a Morbidly Obese Population: A Neglected Trinomial"

_nutrients, 2022, doi:10.3390/nu14153211_

Round 1

Reviewer 1 Report

Caudet et al. focused their studies on the interactions between prokaryotic and eukaryotic intestinal communities when correlated with metabolic comorbidities and diet. The paper is within the scope of the journal and the topic is interesting and relevant for the field. The research is well described, even if an extensive revision of the text editing is necessary, and tables and figures are appropriate. The manuscript is well organized, readily understandable and the research of information has been accurate, with updated references.

Author Response

We do really appreciate the reviewer’s comments, which have improved our manuscript.

Below, please find the provided answers:

Comments and Suggestions for Authors

Caudet et al. focused their studies on the interactions between prokaryotic and eukaryotic intestinal communities when correlated with metabolic comorbidities and diet. The paper is within the scope of the journal and the topic is interesting and relevant for the field. The research is well described, even if an extensive revision of the text editing is necessary, and tables and figures are appropriate. The manuscript is well organized, readily understandable and the research of information has been accurate, with updated references.

The edition of the text, as well as the English, have been revised in the new version of the manuscript.

Reviewer 2 Report

This is a comprehensive and well designed study investigating gut microbiota in a morbidly obese subjects demonstrating differencies in functioning microbiota dependant metabolic syndrome, insuilin resistance, NAFLD, diabetes type 2-

However, subgropus of patients are relatively small to draw definite consclusions, and statistical methodology used allows only to determine differencijes between gropus and not correlation of mentioned differencies and particular comorbidity.

Therefore the consclusion can not be "The presence of 611 Blastocystis sp. and/or Dientamoeba fragilis correlates with higher bacterial richness and 612 with a trend towards higher bacterial diversity." Please rewrite every other section mentioning correlation or implement statistical methods for correlation analysis.

In addition logistic regression analysis could be implemented to determine relationship between colonization (yes and no) and other confounding factors such as age, gender, metabolic comorbidities, 260 meal replacement consumption and smoking condition, obesity, DMT2, insulin resistance, ...

Author Response

Comments and Suggestions for Authors

This is a comprehensive and well designed study investigating gut microbiota in a morbidly obese subjects demonstrating differencies in functioning microbiota dependant metabolic syndrome, insuilin resistance, NAFLD, diabetes type 2-

However, subgropus of patients are relatively small to draw definite consclusions, and statistical methodology used allows only to determine differencijes between gropus and not correlation of mentioned differencies and particular comorbidity.

  1. Therefore the consclusion can not be "The presence of 611 Blastocystis sp. and/or Dientamoeba fragilis correlates with higher bacterial richness and 612 with a trend towards higher bacterial diversity." Please rewrite every other section mentioning correlation or implement statistical methods for correlation analysis.

We agree with the reviewer and appreciate his/her comment. The word "correlates" is not appropriate to comment on the differences found, taking into account the statistical analyses carried out, which have consisted of comparing the mean value of each of the indexes (alpha and beta) between groups. The text of the manuscript has been adequate and the word "correlates" has been appropriately replaced.

  1. In addition logistic regression analysis could be implemented to determine relationship between colonization (yes and no) and other confounding factors such as age, gender, metabolic comorbidities, 260 meal replacement consumption and smoking condition, obesity, DMT2, insulin resistance, ...

Once again we agree with the referee. A binary logistic regression analysis could yield more information on the association between variables, but, at this time, the biostatistical analysis of the results is already closed and we do not have the possibility to do it. However, we will extend the statistical analysis, including the proposed BLR, in future studies.

Round 2

Reviewer 2 Report

The manuscript can be accepted in the present form.